# Reversal of the Word Sense Disambiguation Task Using a Deep Learning Model

Algirdas Laukaitis

The Faculty of Fundamental Sciences, Vilnius Gediminas Technical University, Saulėtekio al. 11,
LT-10223 Vilnius, Lithuania; algirdas.laukaitis@vilniustech.lt

**Abstract:** Word sense disambiguation (WSD) remains a persistent challenge in the natural language processing (NLP) community. While various NLP packages exist, the Lesk algorithm in the NLTK library demonstrates suboptimal accuracy. In this research article, we propose an innovative methodology and an open-source framework that effectively addresses the challenges of WSD by optimizing memory usage without compromising accuracy. Our system seamlessly integrates WSD into NLP tasks, offering functionality similar to that provided by the NLTK library. However, we go beyond the existing approaches by introducing a novel idea related to WSD. Specifically, we leverage deep neural networks and consider the language patterns learned by these models as the new gold standard. This approach suggests modifying existing semantic dictionaries, such as WordNet, to align with these patterns. Empirical validation through a series of experiments confirmed the effectiveness of our proposed method, achieving state-of-the-art performance across multiple WSD datasets. Notably, our system does not require the installation of additional software beyond the well-known Python libraries. The classification model is saved in a readily usable text format, and the entire framework (model and data) is publicly available on GitHub for the NLP research community.

**Keywords:** word sense disambiguation; natural language processing; WordNet



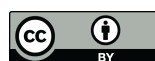

## 1. Introduction

The task of determining the semantic meaning of a word from the context of a sentence is a challenging problem in natural language processing (NLP) that can be difficult for both humans and computers [1]. The word sense disambiguation (WSD) problem is one such challenge that has been the subject of extensive research in the field of NLP. In fact, some researchers believe that the WSD problem is AI-complete [2]. This means that if a computer could solve this problem, it would be able to solve all other problems of artificial intelligence at the same time. However, in this study, the WSD problem was treated as a simple data classification problem, which large neural networks must be able to solve with almost 100% accuracy if a suitable classification scheme is provided.

Various WSD solutions have been proposed over several decades of research, among which, deep neural networks are the most effective [3]. These solutions model the WSD problem as a classification task, where each class corresponds to a specific concept or synset. Usually, a separate model is trained for each word on a corpus of sentences annotated with that word. However, a neural network-based WSD solution requires a large amount of annotated data, which may not be available for many words. Therefore, knowledge-based methods, which exploit the relationships between semantic categories in a knowledge base, offer an alternative way to solve the WSD problem [4,5]. These methods infer the most likely sense of a word on the basis of the semantic context.

One of the main assumptions of knowledge-based methods is that the relationships between concepts in the knowledge base are accurate and reliable, i.e., they are free of errors. To illustrate that this is not always the case, we took the word 'table' from the

semantic dictionary WordNet [5], which was the primary knowledge base used in this study, and examined its suitability as a formal ontology.

The word 'table' has eight senses, six of which are nouns. Only two of these noun senses can be considered as ontology classes: "table%1:14:00::" (a set of data arranged in rows and columns) and "table%1:06:01::" (a piece of furniture having a smooth flat top that is usually supported by one or more vertical legs). The remaining four noun senses are not appropriate as ontology classes of the word "table". For instance, "table%1:06:02::" (a piece of furniture with tableware for a meal laid out on it) is a specific instance of the sense "table%1:06:01::" and can be ignored for many semantic analysis tasks of natural language. Another sense, "table%1:14:01::" (a company of people assembled at a table for a meal or game), is described in WordNet as an abstract entity and it is impossible logically derive from the WordNet knowledge base that it is a collection of physical entities.

Some may contest the assertion that the word "table" possesses only two discernible semantic meanings, thereby raising the query whether a formal methodology can be devised for categorizing words into distinct semantic classes. This question becomes even more pertinent when considering the lack of uniformity in the existing approaches. Previous works [6–9] demonstrated a significant disparity in how semantic meanings are classified for individual words. This inconsistency poses a challenge for researchers attempting to compare findings across different studies and hinders the development of a unified framework for semantic analysis.

Addressing this query constitutes a focal point of this research article. The central idea of our proposed formalization revolves around harnessing the capabilities of deep learning models to extract word embedding vectors. It is assumed that these vectors, when subjected to clustering analysis, should correspond closely to the conceptual categories delineated within a formal ontology, given sufficient training and parameterization of the deep learning model. Figure 1 illustrates this premise, depicting the projection of embedded vectors onto the first two principal components derived through principal component analysis (PCA), obtained from the BERT [10] model after processing sentences containing the term "table". Notably, the visual representation in the figure reveals two distinct clusters discerned by the BERT model. The method presented within this study facilitated the consolidation of six synsets attributed to the word "table" within WordNet into two coherent categories. Remarkably, the alignment between these two synsets and the delineated clusters in the PCA projection underscored the efficacy of the approach presented here in aligning the semantic categorization with the inherent structure of the deep learning model's representations.

To validate the alignment between our proposed deep learning-based categorization and established ontological structures, we used a custom WSD debugging system. This system, detailed in Section 4, facilitates the systematic analysis of erroneous classification results. It allows for the exploration of alternative WordNet synset labels and their impact on the outcome of categorization. This iterative process enables the identification of synset groupings that yield more semantically consistent results, aligning with the inherent structure of the deep learning model's representations. Figure 1 provides a visual representation of our findings for the word "table", showcasing two distinct images obtained during the evaluation process. On the left side of Figure 1, we observe the image generated when all the WordNet synset labels remain unchanged. Conversely, the image on the right depicts the scenario where specific synset labels, namely "table%1:06:02::" and "table%1:14:01::", have been modified to the label "table%1:06:01::". These adjustments were made after an analysis using the WSD debugging system.

The WSD debugging system played a crucial role in identifying erroneous classification results. By systematically examining the synsets' values, we were able to justify the combination of certain synsets. Notably, this formal justification enhanced the reliability of our approach and contributed to the robustness of our WSD model. For further details and technical specifics regarding the WSD debugging system, interested readers can refer to the GitHub repository associated with this project (Supplementary Materials).

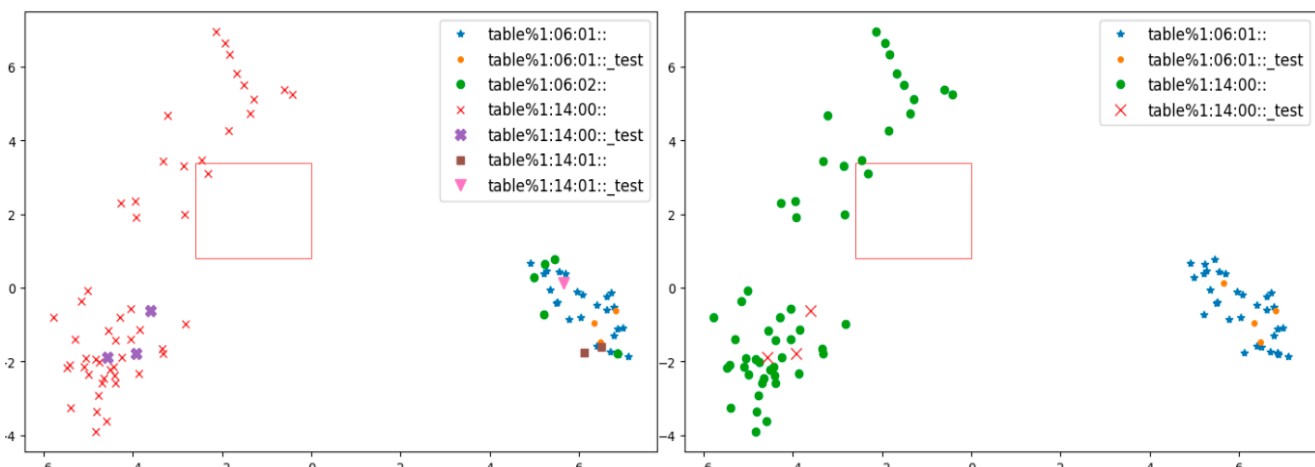

**Figure 1.** Clusters of the word "table" embedded in the WordNet corpus. The projection of embedded vectors is onto the first two principal components. The images were obtained using the WSD debugging system, which is presented in Section 4 of this study. The (**left**) side is the image when all of the WordNet synset labels are left without changing them. The (**right**) is the image when the synset labels "table%1:06:02::" and "table%1:14:01::" are changed to the label "table%1:06:01::".

Another important assumption made in WSD tasks is the assumption that human-annotated sentences are the gold standard that is error-free. However, this assumption is fundamentally flawed, as errors can occur not only in programming but also in the process of annotating data. During programming, we can compile and test the program that is to eliminate some errors. Meanwhile, when annotating natural language sentences, we often have no feedback about the quality of the annotations and, as a result, the probability of errors is quite high.

The method presented in this study abandoned these two assumptions and considered that the uncertainty of the WSD problem arises from the following elements of the knowledge base: (1) not having enough annotated sentences; (2) poorly annotated sentences, that is, cases when the person annotating the sentences made a mistake in determining the semantic meaning of the word; and (3) an incomplete and erroneously designed knowledge base. By addressing these issues, the suggested method is able to achieve high accuracy in solving the WSD task.

In addition to the proposed WSD method, this study introduces a system that effectively implements it. The system consists of two essential components, the first of which is the novel WSD system, characterized by its ability to achieve a classification accuracy comparable with state-of-the-art systems while maintaining relatively low consumption of computational memory. This makes it particularly valuable in natural language tasks that involve the use of embedded word vectors derived from deep neural networks, where computational resources are often limited.

The other component of the system is a debugging tool for the WSD task. The key idea behind this tool is to use embedded vector clusters as the reference standard, allowing the detection of outliers in the embedding space that indicate inconsistent annotations or inaccuracies in the design of the knowledge base. The motivation for developing this tool came from the system's error analysis. Interestingly, the analysis showed that many classification errors were due to the human annotators' mistakes rather than the system's limitations.

The rest of this article is organized as follows. Section 2 describes the new WSD system, which has two main advantages over other WSD systems. First, it achieves comparable accuracy while consuming much less computational memory. Second, it is implemented in a Google Colab notebook, which enables easy testing and integration of the system with other NLP frameworks using only a web browser.

Section 3 introduces the system's training algorithm, highlighting its reliance on standard machine learning libraries, a unique feature that enhances its accessibility and extensibility. Section 4 introduces the debugging method of the WSD task, a tool for creating semantic dictionaries and ontologies. This WSD debugging method not only identifies incorrectly annotated sentences but also lets users refine and correct these sentences, assessing their impact on the model's training. Section 6 presents the results of a comprehensive experiment, encompassing standard datasets for the WSD problem, to evaluate the developed system's accuracy. The article concludes with a summary and a discussion of future work.

## 2. The WSD Framework

The proposed word sense disambiguation framework sits within a broader natural language processing pipeline (Figure 2). This minimal representation showcases the system's integration and clarifies its role in addressing the WSD problem. The framework comprises eight key components. To ensure reproducibility, we have developed a comprehensive Google Colab notebook that mirrors the step-by-step process depicted in Figure 2. This Colab notebook serves as a practical guide for researchers and practitioners interested in implementing the proposed method. Readers can access the Colab notebook via the GitHub repository associated with this project (Supplementary Materials). The Colab notebook is designed for simplicity and ease of use. By executing a single command ("Run All"), users can seamlessly reproduce our entire methodology. In the following description, we provide a detailed breakdown of each of the eight essential steps involved in this word sense disambiguation framework:

1.  Your text: The initial stage begins by receiving raw text input.
2.  spaCy NLP: This component processes the text, extracting word lemmas and determining grammatical forms through spaCy's NLP models [11].
3.  KerasNLP BERT: Leveraging the KerasNLP library [12], the system generates contextual BERT neural network embedding vectors for each word in the sentence.
4.  Word alignment: This alignment results from a combination of both the spaCy and KerasNLP models, creating an index that maps corresponding words between the two representations.
5.  NLTK WordNet: Using the NLTK [13] WordNet synset tags, this module annotates all the words in the text. The algorithm selects the most frequent synset for each word.
6.  NCA projections: The system projects each word into an n-dimensional space derived from the training data, capturing the essential semantic relationships.
7.  kNN classification: The kNN method is used for the classification of words that potentially have multiple semantic meanings, where their lemma can be described by several WordNet synset values.
8.  Final classification: This model consolidates the information from previous stages, providing the final, refined classification for each word.

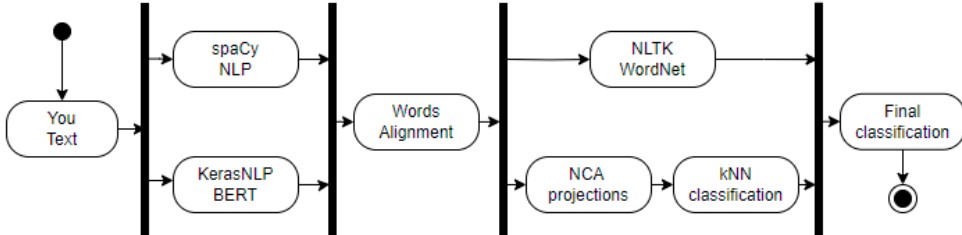

**Figure 2.** A general implementation of the WSD method.

Detailed descriptions of the critical components follow, offering a deeper understanding of the WSD framework's inner workings.

**spaCy NLP.** The core component of the proposed WSD framework is the spaCy NLP library. spaCy offers several functionalities that are critical to WSD, each addressing a specific aspect of the problem, as follows.

- **Tokenization and lemmatization.** spaCy segments the input text into individual words (tokens) and identifies the lemma (base form) for each token. This is crucial for WSD, as the synset classifier operates on lemmas, ensuring consistent representation across different inflections of the word.
- **Noun phrase chunking.** spaCy identifies noun phrases (NPs) within the text. NPs play a twofold role. First, they help pinpoint potential word combinations aligning with distinct synsets in WordNet. Second, chunking facilitates the identification of the head noun within the NP, enabling the selection of the most relevant synset for the entire phrase.
- **Part-of-speech (POS) tagging.** spaCy assigns grammatical tags (POS tags) to each word, indicating its role within the sentence (e.g., noun, verb, adjective). POS tags are valuable features for the synset classifier, providing additional context alongside the lemma for more accurate WSD.
- **Dependency parsing.** spaCy can also extract the dependency relationships between words in a sentence. While not currently utilized in this WSD framework, dependency information holds promise for future development. Analyzing these relationships could refine word sense disambiguation by considering the syntactic structure of the sentence.
- **Named entity recognition (NER):** NLP libraries often include NER modules that identify predefined named entities such as people, locations, and organizations. However, NER typically covers a limited range of semantic categories compared with WSD. In spaCy and similar libraries (e.g., Stanford CoreNLP [14]), NER addresses only a subset of noun phrases. This framework proposes aligning the NER categories with their corresponding hypernyms in WordNet, potentially leveraging the existing NER functionalities to support WSD. By contrast, the WordNet WSD module independently identifies words and phrases that are suitable for disambiguation, going beyond the limitations of NER.

Two widely recognized NLP libraries, spaCy and Stanford CoreNLP, were subjected to experimentation. Individual Colab documents were generated for each library, presenting a comprehensive illustration of their potential in addressing the problem of word sense disambiguation. The subsequent discussion centers on the spaCy library, with the understanding that the outlined principles are equally applicable to the Stanford CoreNLP library.

**KerasNLP BERT.** A crucial element of the proposed WSD method is the integration of the KerasNLP. This library offers the capability to leverage various pretrained deep neural network models, enabling further training on fresh data or the extraction of embedded word vectors. In the context of this research, the focus was on showcasing the outcomes derived from using the BERT neural network. BERT's adeptness lies in its capacity to generate embedding vectors for words within a sentence, a process that is integral to assessing the significance of each word synset. By using BERT's word embedding vectors, we aimed to enhance the precision and granularity of the results of disambiguation, thereby reinforcing the overall efficacy of our WSD approach.

**Word alignment.** The integration of the spaCy NLP and KerasNLP modules stands as a pivotal aspect of our methodology. These modules operate independently, each generating its own tables of words (or tokens, in the case of KerasNLP). However, due to potential disparities between these tables, an additional step is necessary to establish correspondence. This entails creating an index that maps spaCy's words to their respective tokens in KerasNLP. The algorithm facilitating this matching process is straightforward and involves the following key steps:

1. Through sequential scanning of the spaCy word table, each word is matched to the beginning of a token from the list generated by BERT's tokenizer.

2.  To identify the corresponding beginning of the word (token) from the BERT module, consecutive tokens are successively combined and compared against the given spaCy word. Upon finding a match, the index of the first located BERT token is returned. This iterative process ensures the alignment of spaCy words with their corresponding tokens in KerasNLP, facilitating integration within the WSD framework.

**NLTK WordNet.** The NLTK WordNet module is a key component of our WSD methodology, as it provides a convenient API for accessing the WordNet dictionary. This module helps us assign synset values to words that either have no sample sentences or have only one synset value. Furthermore, the choice of the NLTK WordNet module was driven by important non-functional requirements that emerged during the project's development.

One of these requirements was the compatibility of the WSD system code with the Google Colab environment, especially when using the "Run all" menu option. Among the various approaches to the WSD problem that have been proposed in the last 30 years, only the NLTK WordNet module, along with its WSD Lesk algorithm [15], fulfilled this requirement. This demonstrates the NLTK WordNet module's dependability and flexibility, making it an essential element of our WSD framework.

**NCA projections.** The Nearest Neighbor (KNN) algorithm is a simple yet powerful method for solving the problem of word sense disambiguation. It only depends on one parameter, and its accuracy improves as more training data become available. However, KNN faces a significant challenge when applied to the WordNet WSD problem. This challenge stems from the high computational cost of storing and searching the whole dataset in memory to find the synsets for each word in a given sentence. In our approach, we overcame this limitation of KNN by using a technique that projects the embedded word vectors into a lower-dimensional subspace before applying the KNN classifier. The dimensionality of this subspace is dynamically adjusted during the training phase. To choose the best components for this transformation, we used the neighborhood components analysis (NCA) method [16], which ensures an efficient and effective solution to the WSD problem within the WordNet framework.

**Final classification.** The problem of word sense disambiguation is often a prerequisite for solving other tasks in natural language processing. The final classification module, which integrates the results of previous steps, is crucial for delivering these results to the specific task that needs a resolution of WSD. In this project, we have identified some NLP tasks that benefit from the outcomes of WSD. One of them is 3D scene generation, where the WSD results help us make two important decisions: determining whether a noun phrase refers to a physical object that needs to be represented in the 3D scene, and finding out whether the objects in the scene have any movement. Another task is animating works of fiction. As in 3D scenes, identifying noun phrases that describe relevant objects is essential. Therefore, the algorithm for the final classification module consists of the following steps:

1.  Using the spaCy library, we extract noun and verb phrases.
2.  Using spaCy's grammatical relation analysis, we find the main (head) word in each phrase.
3.  Using the WSD module, we select noun phrases, where the head word indicates a physical object. We also mark verb phrases where the head word expresses movement of a physical body.
4.  Only the selected phrases that meet these criteria are used for the final NLP task, which involves phrases related to physical objects and stages of movement, thus enhancing the efficiency and effectiveness of subsequent NLP operations.

## 3. Training

The principal objective of this project was to develop a model capable of achieving accuracy comparable with existing models utilizing embedded word vectors, as demonstrated in [17–19]. A key innovation pursued from inception was the imperative for the model to significantly reduce its computational memory footprint while maintaining such levels of accuracy.

Following the successful achievement of predefined targets related to the model's size and accuracy, an exhaustive investigation was undertaken to dissect the underlying causes of error propagation in the context of word sense disambiguation. Leveraging established datasets [17,20–23], our analysis revealed that certain inaccuracies did not originate from inherent deficiencies in the model but rather from distinct sources, as follows.

1.  **Discrepancies in the test dataset:** The first source of error was discrepancies within the test datasets themselves. Variability in the quality of annotation, the distribution of the data, and contextual diversity posed challenges for accurate disambiguation. In Figure 3 (left), we observe a sample of test data associated with the synset value "level%1:26:00::_test", which exhibits proximity to the group of training data represented by the synset value "level%1:07:00::". Further investigation using tools from the WSD debugging system revealed that this discrepancy arose from an annotation error in the standard test data [23]. Specifically, the sentence with the identifier "semeval2013.d000.s008" contained the phrase "emissions . . . compared with 1990 levels". Here, the model identified a possible misinterpretation of "levels" and assigned it the sense of "a position on a scale of intensity or amount or quality" instead of "a specific identifiable position in a continuum or series or especially in a process". The analyst reviewed the model's proposed corrections and, if in agreement, incorporated them into the "sentence_synset_to_synset.tsv" corrections file (available on the project's GitHub page). On the right side of Figure 3, we present an image reflecting these corrected annotations.

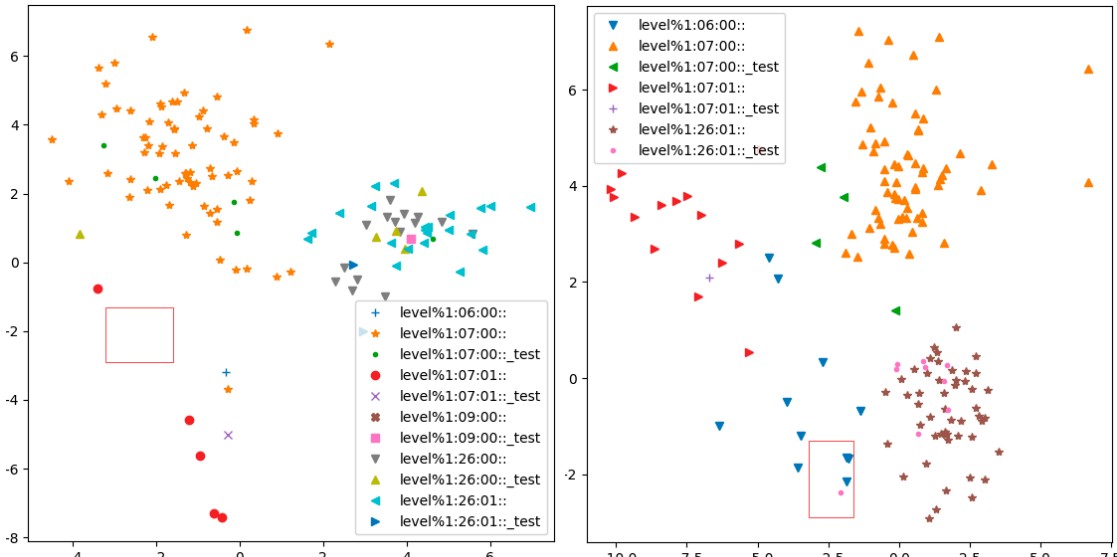

**Figure 3.** This figure showcases the impact of the proposed data transformation methodology on word sense disambiguation. Each point on the (**left**) side represents a sentence containing the noun "level" from the standard WordNet dataset. The (**right**) side depicts the corresponding data points after applying the transformations outlined in the study. By visually comparing the points, we can observe how the transformations effectively separate data points belonging to different senses of the word "level", improving the model's ability to distinguish between them during disambiguation.

2.  **Annotation errors in the training data:** Secondly, we identified annotation errors within the training dataset [24]. These inaccuracies were propagated through the model, affecting its performance during inference. Addressing and rectifying such errors became crucial for enhancing the model's robustness. Similar to the test dataset, the training data also exhibited annotation errors. We identified these errors by visually inspecting NCA projections of the data points. Points that deviated significantly from their designated synset cluster were flagged for potential misannotation. Fol-

lowing the analyst's verification of the context and intended sense, corrections were incorporated into the "sentence_synset_to_synset.tsv" file for retraining the model.

3. **Limitations of WordNet synsets:** Our investigation highlighted inadequacies within WordNet synsets' relationships and descriptions. While WordNet serves as a valuable lexical resource, it occasionally fails to capture subtle nuances required for comprehensive word sense disambiguation. Notably, we observed an overly granular segmentation of certain words into synsets, hindering accurate resolution of their sense. Therefore, when modifying WordNet synset lists or relations, it is crucial to consider the intended application of natural language processing. In this work, synset merging decisions were guided by the ultimate goal of using NLP to generate 2D and 3D models, as described in [1]. Over the past five decades, starting with the pioneering SHRDLU system [25], numerous systems have attempted natural language manipulation of objects of computer graphics (see [26] for a review of 26 such systems). Many of these systems process multiple sentences to identify physical objects within a 3D scene, often leveraging spatial knowledge to resolve ambiguities (e.g., SceneSeer [27]). This specific NLP application guided the WSD solution presented in this study. Returning to the example depicted in Figure 3, we made the decision to merge the synsets "level%1:26:00::", "level%1:09:00::", and "level%1:26:01::" into a single group. Subsequently, Figure 3 (right) illustrates how the training sample points align in the plane defined by the two NCA components. Throughout the course of this WSD project, approximately 1800 such changes were implemented, and all details are accessible on the project's GitHub repository in the file "synset_to_synset.tsv".

4. **Scarcity of Data for Specific Words:** Another critical constraint emerged: the lack of sufficient data to construct reliable classifiers for specific words. This limitation hindered the model's ability to generalize effectively across the entire lexicon, particularly for low-frequency or domain-specific terms. Returning to the example depicted in Figure 3 (left), we observe that the synset value "level%1:06:00::" is associated with only one training sentence and one test sentence. These sentences are significantly distant from each other in the NCA component space, resulting in model errors. To address this challenge, we generated sentences using ChatGPT or Gemini AI agents specifically for synset values with limited examples. These AI agents were prompted with descriptions of synsets from the WordNet dictionary. Notably, on the right side of Figure 3, we observe that this approach proved beneficial: the sentences for the synset value "level%1:06:00::" formed a distinct cluster, and the test sentence aligned within this cluster.

To address the limitations identified in the error analysis, we propose the refined model training process outlined in Figure 4. This section delves into a detailed breakdown of each component and its contribution to the enhanced performance of the model.

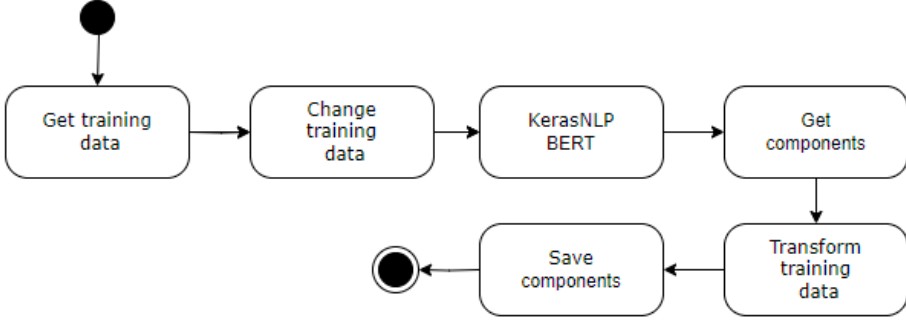

**Figure 4.** Process of the WSD training model.

**Obtaining the training data.** This project used a multi-phased approach to acquiring data, aiming to create a robust and iteratively improvable WSD model.

The initial training data consisted of sentences from the widely used SemCor corpus [24]. While effective, this dataset's limitations necessitated exploring alternative sources.

Here, we detail the exploration of various data augmentation techniques and their impact on the model's performance.

WordNet definitions: Initially, we attempted to improve the model's performance by augmenting data using the existing WordNet synsets' definitions. However, utilizing the WSD debugging method described later, we observed a detrimental effect on accuracy when incorporating certain definitions. Consequently, some definitions were excluded during the process of iterative improvement.

Artificially generated sentences: We further investigated the inclusion of artificially generated sentences based on WordNet's hypernym and hyponym relations. These sentences were constructed by combining hypernym/hyponym definitions with a phrase indicating the intended meaning. Evaluation revealed this approach to be ineffective, and the generated sentences were not used in further training.

Wikipedia articles: Given the BERT neural network's pretraining on Wikipedia data, we explored the impact of incorporating additional Wikipedia sentences. This approach proved successful. Adding approximately 10 new sentences per novel synset value demonstrably improved the model's ability to identify those synsets.

FrameNet sentences: Considering the planned future integration of FrameNet with WordNet, we investigated the inclusion of FrameNet sentences. While a comprehensive FrameNet–WordNet merger falls outside this study's scope, a subset of FrameNet sentences was incorporated to enhance the model's performance.

Chatbot-generated sentences: Finally, we explored the use of chatbots (ChatGPT 3.5, Microsoft Copilot, and Google Gemini) for generating new training data. This method offered a convenient way to obtain novel sentences. However, rigorous human expert evaluation ensured that only accurate sentences were added to the training data.

**Changing the training data.** Following acquisition of the data, a process of data replacement ensued, whereby synset tags of certain words were substituted either due to mislabeling within the dataset or as a result of the consolidation of WordNet synsets by the knowledge base manager, thereby creating a new version of WordNet. All requisite transformations were documented in textual format on the GitHub portal, facilitating direct editing through the platform's integrated text editor.

**KerasNLP BERT.** Subsequently, the KerasNLP library was employed to generate embedding vectors for each word within the sentences. Noteworthy observations during this phase highlighted a limitation of the BERT model pertaining to its sensitivity when encountering words absent from its dictionary. For instance, the word "Fujimoto" would be tokenized into "Fuji", "mo", and "to", potentially leading to significant deviations in the embedded vectors for other words within the sentence. Nevertheless, the selection of the KerasNLP library was predicated upon its adherence to key non-functional requirements, namely stable performance, rapid loading of the model, and the anticipated long-term support.

**Obtaining the components.** This project explored a novel approach to word sense disambiguation (WSD) by leveraging neighborhood component analysis (NCA) in conjunction with word embedding vectors derived from the BERT neural network. While established methods such as fine-tuning and transfer learning excel at integrating pretrained networks for new tasks (e.g., sentiment analysis), they are often tailored towards creating new classifiers. Our objective, however, diverged from simply generating a WordNet synset classifier. Instead, we aimed to reshape WordNet's structure to align better with the knowledge encoded within BERT. This focus on interactive restructuring of knowledge necessitated a WSD method that facilitates the visualization of learned knowledge.

Here, word embedding vectors and their multidimensional projections became the cornerstone of selecting the model due to their inherent visualizability. Following a comprehensive evaluation of the vector design methods offered by scikit-learn, NCA projections were chosen for their demonstrably superior performance in identifying the synset values.

We further refined the NCA model by meticulously tuning its meta-parameters. Notably, each word sense was assigned a k-nearest neighbors (k-NN) classifier operating

within the n-dimensional NCA component space. Exploration of these parameters (k ranging from 2 to 5, n ranging from 2 to 5) was conducted during the model's training, with a focus on maximizing the classification accuracy. Additionally, a stringent regularization criterion was implemented: each unit of increase in a parameter had to yield at least a 1% enhancement of the model's accuracy. This ensured robustness and stability during the training process.

**Transforming the training data.** In our model, we used a systematic approach to project word embedding vectors into the foundational NCA component space. This projection process constitutes the core architecture of our model, which is composed of two principal elements: the NCA component vectors and the training data projections derived from the NCA framework. It is crucial to highlight that both vector types (the NCA components and the data projections) were precisely rounded to two decimal places. This exacting standard served a twofold purpose: it optimizes the utilization of memory resource, facilitating efficient data storage and computational processing, and it ensures uniformity and coherence in the model's representational schema. By adopting this method, we achieved equilibrium between computational efficiency and the fidelity of the model's representations, allowing for smooth integration and deployment across various computational settings.

**Saving the components.** A critical stage in the training process involves the development of a dedicated classifier. This independent classifier plays a vital role in refining the model's capacity for accurate word sense disambiguation. Once the training phase is complete, the WSD model is serialized and stored in the JSON format. JSON (JavaScript Object Notation) is a well-established and interoperable data exchange standard that facilitates the efficient storage and transmission of the model's architecture and parameters. This preservation strategy ensures seamless accessibility and enables straightforward dissemination of the model for future research applications.

## 4. Debugging and the Knowledge Base Management System

The WSD model relies heavily on established lexical resources such as WordNet and the SemCor corpus. While these resources have proven valuable for NLP tasks, their lack of recent updates (over a decade) can limit their ability to handle the evolution of language and the emergence of new word senses. This study proposes a systematic approach to address this limitation. We aimed to enhance the quality and coverage of WordNet and SemCor for WSD tasks. This section will focus on debugging and expanding the knowledge base within WordNet, as well as enriching the annotated textual sentences in SemCor. By improving these resources, we aimed to increase the adaptability and applicability of WordNet to diverse linguistic contexts and domains.

This study proposes a multifaceted methodology to address the limitations in existing lexical resources such as WordNet. The methodology focuses on two key aspects: (1) debugging the WordNet knowledge base to identify and rectify inconsistencies and inaccuracies in the synsets' descriptions, thereby enhancing their reliability for WSD tasks; (2) expanding WordNet with new synsets to accommodate the evolving nature of language and ensure its continued relevance for contemporary NLP applications. By improving the quality and coverage of WordNet and similar resources such as SemCor, this approach aimed to increase the adaptability and applicability of these tools to diverse linguistic contexts and domains.

Figure 5 illustrates a schematic of the proposed WSD debugging methodology, which is a five-stage process designed to address inconsistencies and inaccuracies in WordNet. The initial stages (1 and 2) focus on preparation of the data and semantic representation. Stage 1 involves rigorous selection of the data and preprocessing of the training and testing data (including cleaning and augmentation) to ensure the data's quality and representativeness. Stage 2 leverages the pretrained "KerasNLP BERT" module to acquire embedded word vectors from the prepared data. This captures rich semantic representations for each word within its context.

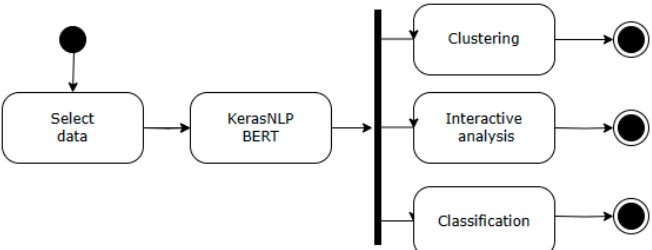

**Figure 5.** WSD debugging method.

Stages 3–5 address exploration and refinement of the knowledge base. Stage 3 performs cluster analysis on the embedded vectors. Principal component analysis (PCA) reduces the dimensionality and groups words with similar semantic properties, facilitating consolidation of the synsets and revealing potential inconsistencies. Stage 4 utilizes projection analysis to provide an interactive interface for exploring the semantic relationships between synsets. This visual exploration (2D/3D) helps identify and rectify inconsistencies within the hierarchy of synsets. Finally, Stage 5 involves the evaluation and integration of various candidate models of WSD. This rigorous assessment ensures the selection of the most effective model for integration, ultimately improving the performance of disambiguation. Next, we describe these steps in the WSD debugging task in more detail.

### 4.1. Select Data

This work introduces the "Select Data" process, a flexible framework for evaluating the word sense disambiguation model's performance through targeted data selection (Figure 6). The process empowers researchers to investigate the influence of both individual words and the characteristics of word group on the model's accuracy.

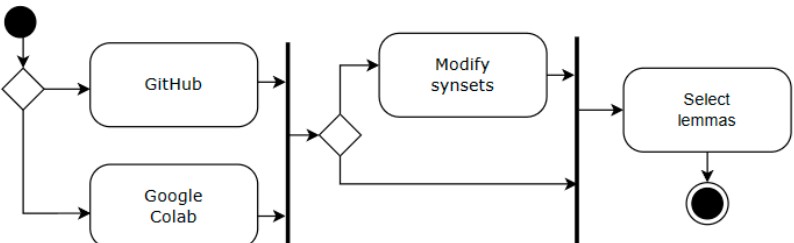

**Figure 6.** The Select Data subprocess.

The first step involves choosing an appropriate data source. This project used a curated training and testing dataset publicly available on GitHub (https://github.com/aalgirdas/wordnet_onto (accessed on 1 June 2024)). Alternatively, researchers can leverage custom datasets stored locally (e.g., Google Drive) to address specific research questions.

Despite its extensive history, WordNet remains a foundational resource for WSD tasks. However, inaccuracies in annotation of the corpus and the WordNet synsets' relationships were identified during this project, necessitating modifications for improved semantic representation. The "Modify Synsets" step allows researchers to select either the original WordNet version or implement targeted modifications of the synsets' descriptions and sentence annotations. Importantly, all modifications have been meticulously documented and stored on GitHub, enabling transparent and reproducible research.

Finally, the "Target Word Selection" step provides granular control over the analysis. Researchers can choose to investigate a specific lemma, select individual words, or designate a list of lemmas for analysis. This flexibility ensures that the investigations align precisely with research objectives, facilitating a more refined and targeted approach to the development of WSD models.

*4.2. Clustering*

This work hypothesized that a deep neural network (DNN) trained on comprehensive natural language processing tasks can implicitly capture semantic relationships that are useful for word sense disambiguation. We propose an approach where the embedded vectors learned by such a DNN are clustered, with each cluster potentially representing a distinct sense. By comparing these data-driven clusters with the existing sense inventory of a knowledge base (e.g., WordNet), discrepancies between the two can be identified. These inconsistencies may indicate the need for refinements of either the knowledge base's sense definitions or the DNN's architecture and training data.

To facilitate WSD debugging and refinement of the knowledge base, we propose a clustering-based approach to analyze the embedded vectors. This method empowers knowledge engineers to make informed decisions regarding the merging, deletion, or creation of synsets. Two clustering algorithms were considered for this purpose: principal component analysis and neighborhood component analysis. While KMeans was used for all clustering tasks throughout this project for consistency, future work will explore the potential benefits of utilizing alternative algorithms in the context of WSD debugging to assess their impact on efficacy.

The initial phase of clustering entails determining the optimal number of clusters. The algorithm unfolds through several key steps:

1. **Selection of the method of vector projection**: The clustering algorithm offers flexibility in the methods of vector projection, including principal component analysis or neighborhood component analysis.
2. **Vector projection**: Embedded vectors undergo projection, a crucial preparatory step for subsequent calculations.
3. **Determination and evaluation of clusters as follows.**
    3.1. Application of the KMeans clustering algorithm: Use of the KMeans algorithm on the projected vectors enables the delineation of clusters and the assignment of each sample to its corresponding cluster.
    3.2. Metric calculation: Two pivotal metrics, namely the silhouette score and the adjusted mutual information score, are computed for each potential configuration of the clusters, providing quantitative insights into the quality and coherence of the resulting clusters.
4. **Selection of the optimal cluster**: The ultimate determination of the ideal number of clusters hinges upon selecting the configuration yielding the highest product of the silhouette score and the adjusted mutual information score, indicative of optimal clustering performance.

This systematic approach to determination of the clusters not only ensures robustness and reproducibility but also facilitates informed decision-making for knowledge engineers engaged in WSD debugging tasks. Furthermore, the incorporation of diverse vector design methods and comprehensive evaluation metrics underscores the methodological rigor that is essential for advancing the field of natural language processing.

*4.3. Clasification*

In alignment with the objectives of this project, the classification module within the WSD debugging task shares a fundamental aim with its clustering counterpart: to equip knowledge engineers with actionable insights for merging, deleting, or introducing new sentence examples. To achieve this objective, a submodel named the "confusion matrix" was designed and implemented. The procedural delineation of this component is outlined as follows.

1. **Partitioning of the data**: The initial step entails determining the allocation ratio for the testing and training data. By default, this parameter is set to 0.1, signifying that 10% of the total dataset will be randomly earmarked for testing purposes, with the remainder designated for training.

2.  **k-Nearest neighbors exploration**: Subsequently, an experimental phase ensues, applying the k-nearest neighbors (KNN) method to the designated testing and training datasets.

3.  **Iterative experimentation**: This experimentation process is iterated N times, facilitating the accumulation of a broader spectrum of measurements.

4.  **Computation of the confusion matrix**: Following experimentation, the confusion matrix is computed on the basis of the amassed data. These results are then presented to the knowledge engineer, who assumes the pivotal role of making the final determinations concerning potential redesigns of the synsets.

This methodical approach to classification not only serves to enhance the efficacy of the WSD debugging endeavor but also empowers knowledge engineers with actionable insights derived from rigorous experimentation. Moreover, the iterative nature of the experimentation process, coupled with the utilization of established methodologies such as KNN and the computation of the confusion matrix, underscores the methodological robustness, which is essential for advancing the domain of natural language processing.

### 4.4. Interactive Analysis for Informed Editing of the Synsets

In tandem with the classification and clustering models, an interactive analysis module was developed to facilitate informed decisions regarding editing of the synsets by knowledge engineers. This module leverages techniques of reducing dimensionality and visual representations to provide a user-friendly exploration of the underlying semantic relationships within the data.

1.  **Visualizing semantic clusters**: The module offers a 2D scatter plot depicting the projections of the words' embedding vectors onto the first two principal components (PCs) derived from the principal component analysis applied to the training data. Projections of the testing data are overlaid for comparison. Users can interactively select data points or regions on the plot to retrieve the corresponding sentence examples. Additionally, the ability to overlay WordNet lexnames, synset labels, or hypernym labels further aids in comprehending whether specific data clusters align with these semantic categories. This visual exploration empowers knowledge engineers to identify potential refinements of the synsets by uncovering clusters that deviate from the expected semantic groupings.

2.  **Reducing the dimensionality and neighborhood analysis**: While PCA provides a valuable 2D visualization for an initial analysis, neighborhood component analysis can be used to assess the information retained in higher dimensions. The NCA projection, presented as another interactive 2D graph, allows users to evaluate the feasibility of disambiguating the target word based on the inherent structure of the data. This visualization can also serve as a valuable tool for identifying potential annotation errors within the training data by highlighting points that deviate significantly from their expected semantic neighbors.

3.  **3D exploration for enhanced insights**: For a more comprehensive perspective, the system offers the option to explore the data in a 3D scatter plot. Similar to the 2D view, users can select between PCA and NCA for reducing the dimensionality, allowing for in-depth examinations of the potential semantic relationships across multiple dimensions. This interactive 3D environment provides additional opportunities for knowledge engineers to refine their understanding of the data and make informed decisions regarding modifications of the synsets.

Overall, the inclusion of the interactive analysis module empowers knowledge engineers with a comprehensive set of visual tools to analyze the semantic relationships within the data [28,29]. This facilitates a more informed approach to editing synsets, ultimately enhancing the effectiveness of the WSD system.

## 5. Results

This work initially focused on enriching the SemCor dataset with WordNet gloss descriptions to enhance semantic representation. The core objective was to develop a WSD model with two key properties: (1) a significantly reduced memory footprint compared with the existing deep learning approaches, ensuring compatibility with resource-constrained environments such as Google Colab; and (2) state-of-the-art accuracy on benchmark datasets such as SemEval. This goal was achieved through a novel method that leverages NCA for reducing the dimensionality and incorporates a vector quantization step, where embedded vector values are rounded to a predefined precision (e.g., two decimal places).

Following the development of the initial model, a detailed error analysis was conducted to investigate the misclassification of a significant number of synsets. This analysis revealed two key issues: (1) the presence of errors in sentence annotations within the training data, and (2) inconsistencies in the synsets' assignments for semantically similar words within WordNet. To address these shortcomings, we developed a WordNet debugging tool specifically designed to identify annotation errors, and proposed modifications for improved alignment between the embedded vector clusters and the WordNet structure. This comprehensive approach not only led to the creation of a novel WSD system but also resulted in the development of an updated WordNet version, ensuring compatibility with the latest iteration. A crucial modification involved the merging of synonymous synsets to maximize the overlap between embedded vector representations and their corresponding WordNet concepts.

To achieve the first project objective of reducing the memory footprint while maintaining accuracy, we investigated techniques of reducing the dimensionality. Notably, the size of the embedded vector base was compressed from 3 gigabytes to 100 megabytes, representing a significant reduction. Interestingly, this compression not only preserved the model's accuracy but also led to a slight improvement in the F1 scores (Table 1).

**Table 1.** Results of WordNet sense disambiguation (F1 score).

| Method | POS | Senseval All |
|---|---|---|
| MFS | Verb | 49.6 |
| | Noun | 67.5 |
| BERT-2NCA 1-NN | Verb | 61.7 |
| | Noun | 74.1 |
| BERT-NCA k-NN | Verb | 64.4 |
| | Noun | 76.7 |
| $BERT_{1024}$ 1-NN | Verb | 63.9 |
| | Noun | 76.4 |

To evaluate the model's performance, we used the Senseval All test dataset [18], encompassing five subdatasets from 2001 to 2015 (Senseval-2 [20], Senseval-3 [21], SemEval-07 [22], SemEval-13 [23], and SemEval-15 [17]). These data are readily available on the project's GitHub repository in a tab-separated values (.tsv) format, where each line represents a synset's test value with its corresponding sentence example.

The training data comprised two components: (1) SemCor [24], a manually sense-annotated corpus containing 226,040 annotations across 352 documents; and (2) WordNet synsets' meanings and definitions, curated by removing outliers identified by the WSD debugging system on the basis of their distance from the class centers in the NCA projection space (details of "bad" definitions are provided in the GitHub repository). Notably, the training data are dynamic and evolve as the WSD model is refined.

This unexpected finding in Table 1 (BERT-NCA k-NN was better than $BERT_{1024}$ 1-NN) suggests the effectiveness of the NCA component model in mitigating noise in the data during the projection of embedding vectors from a high-dimensional space (1024 dimensions) to a lower-dimensional space (ranging from two to five dimensions). Table 1 summarizes

the F1 scores achieved by various models using the unmodified relationships of WordNet synsets. The training data excluded chart-bot generated examples of synsets in order to obtain consistency with the reference models (Most Frequent Sense (MFS) [18] and $BERT_{1024}$ 1-NN [1]).

Additionally, it is essential to discuss the $BERT_{1024}$ 1-NN model, which played a pivotal role in our research. The $BERT_{1024}$ 1-NN model, which uses a BERT neural network [10], generates 1024-dimensional embedding vectors for each word in a sentence (using the first token's vector if a word is split). Word sense disambiguation is achieved by finding the closest training data vector to the new word's vector and assigning its corresponding synset value. This model's large size (>3 GB) limited its deployment on GitHub and motivated the development of the solution presented in this project.

The evaluation in Table 1 focuses solely on nouns and verbs, aligning with the word forms used in the reference work [1]. While the BERT-2NCA 1-NN model, using two fixed NCA components and a kNN classification with k = 1, exhibited a slightly lower F1 score compared with $BERT_{1024}$ 1-NN; the BERT-NCA k-NN model achieved a marginally higher F1 score. This improvement stemmed from its ability to dynamically determine the optimal number of NCA components and the k parameter for each lemma individually. This data-driven approach highlights the efficacy of dynamic parameter tuning in achieving superior performance for semantic classification tasks.

Table 1 clearly demonstrates a significant improvement in the accuracy of noun classification for the WSD task when using neural networks such as BERT. Compared with the MFS baseline, which assigns the most frequent sense in the SemCor dataset to a word, BERT achieved an approximate 10% improvement in the F1 score (Table 1). This finding highlights the limitations of simple rule-based approaches and the effectiveness of deep learning for WSD.

However, the observed improvement of 10% served as a basis for further exploration. We aimed to redefine the state of the art for WSD by pushing the boundaries of accuracy beyond this initial gain. This work further sought to transform WordNet into a formal ontology similar to established resources such as Cyc [30] or SUMO [31,32]. This led to a two-pronged strategy: (1) enhancing the semantic knowledge base (WordNet) itself, and (2) enriching its training corpus with meticulously annotated sentences. The ultimate objective of this approach was to achieve near-perfect WSD accuracy. We implemented three distinct types of modifications to WordNet during the project.

1.  **Refinement of the synsets**: To enhance the accuracy of sense representation, our system implements refinement of the synset. In specific cases, the system automatically replaces synsets with more contextually appropriate alternatives. For instance, in Figure 3, all training and testing examples containing words with the synset value level%1:26:00:: were automatically replaced with level%1:26:01::. The rationale for this replacement is detailed in Section 3. Notably, the system prioritizes user control, allowing easy modification or removal of these replacements through the synset_to_synset.tsv file.

2.  **Pruning**: In the synset_to_synset.tsv file, we conveniently specify the DELETE keyword for synset values we wish to discard. These values correspond to synsets that have been identified as having negligible semantic value, especially those associated with niche domains.

3.  **Expansion of the lexicon**: Novel synsets were introduced to broaden the scope and capture the evolving nature of language. Consider the word "window" as an example. While WordNet lacks a semantic meaning specifically denoting a user interface element in software systems, we addressed this gap by introducing new synset values. To achieve this, we updated the project file NewSynsets.tsv by adding the word "window". We provided a web address (such as a Wikipedia article) where information about this value can be found. Additionally, we specified the synset value from the unmodified (old) WordNet that served as the hypernym (in this case, "window" corresponds to "software%1:10:00::"). Our new WSD system then associates the word

"window" with the synset tag "software%1:10:00::" in relevant sentences. The use of hypernyms ensures compatibility with other WordNet systems, such as NLTK.

It is important to note that the deletions and additions of synsets were primarily undertaken as an experimental investigation, paving the way for future advancements in research into the knowledge base. Table 2 details the statistics of these modifications to WordNet.

**Table 2.** Statistics of WordNet modifications.

| WordNet Modification | Number of Records |
|---|---|
| Synset to synset | 1452 |
| Delete synset | 335 |
| New synset | 7 |
| New sentences from FrameNet | 3091 |
| ChatGPT sentences | 4355 |
| Training sentence corrections | 579 |
| Testing sentence corrections | 126 |

Within the framework of the WSD task, additional refinements were implemented, including the generation of new sentences and rectification of errors in the existing annotations. A detailed breakdown of these modifications is presented in Table 2. It is crucial to recognize the iterative process of development that was undertaken, similar to that of evolving software ecosystems. Each subsequent version of the WSD system incorporated updates to improve the functionality and address new requirements. The statistics provided in Table 2 reflect the version used for the model's classification accuracy reported in Table 3. This highlights the ongoing effort to continually refine WSD capabilities.

**Table 3.** Results of sense disambiguation with WordNet (F1 score).

| Method | POS | Senseval All |
|---|---|---|
| BERT-1NCA 1-NN | Verb | 83.2 |
| | Noun | 95.1 |
| BERT-NCA k-NN | Verb | 84.6 |
| | Noun | 96.7 |
| BERT$_{1024}$ 1-NN | Verb | 83.3 |
| | Noun | 94.4 |

Our work aimed to significantly advance the state of the art in WSD, potentially limiting direct comparisons with previous studies due to the implemented modifications. Table 3 details the accuracy achieved by the model, reflecting these adjustments. While the results exceeded a 92% accuracy threshold, we acknowledge the ongoing pursuit of 100% accuracy. Several factors hindered this goal.

**Limitations in the vVocabulary**: The BERT neural network's inherent limitations in the vocabulary (30,522 words/tokens) can significantly impact WSD outcomes for out-of-vocabulary words.

**Incomplete WordNet modification**: Incomplete supplementation of the data or augmentation of the relationships within the WordNet synsets posed a further challenge.

**The network's Architecture and training data**: The specific architecture of the neural network and the composition of its training dataset likely contributed to the deviation from perfect accuracy. Exhaustive testing was conducted using all variants of the KerasNLP library's neural network, with BERT chosen for comparability with prior research. However, larger LLM networks might yield superior results, warranting future investigation.

## 6. Related Work

Over the years, significant effort has been devoted to developing algorithms and systems for word sense disambiguation since its inception. Among the earliest and most intuitively comprehensible approaches is the Lesk algorithm [15]. Essentially, the Lesk algorithm counts the number of overlapping words between the target word in a sentence and the definition of the synset associated with that word. To address the limitations arising from the Lesk algorithm's heavy reliance on dictionary definitions of the synsets, several extensions have been proposed [33]. Recent advancements have focused on learning distributional word representations from large corpora and replacing the traditional bag of words' overlap with a more sophisticated cosine similarity of the vectors [34].

In parallel, knowledge-based approaches have explored graph theory algorithms to leverage the semantic relationships provided by dictionaries. For instance, the UKB algorithm uses random walks on the semantic graph derived from WordNet to determine word senses using the PageRank metric [35].

Alternatively, machine learning algorithms that construct classifiers from annotated text data have gained prominence. Early works in this field experimented with well-known algorithms such as decision trees [36], support vector machines [37], and neural networks [38].

Numerous WSD systems have been developed to implement these algorithms. In our investigation, we explored systems that are accessible on GitHub. One such system, ConSeC [39], leverages deep neural networks and achieved higher accuracy than the model presented in this article. However, ConSeC requires downloading a large 5 GB model, which is significantly larger than the model discussed here (~100 MB). Additionally, the installation process can be complex (we provide a link to the Colab notebook attempting to run ConSeC in the GitHub repository).

Another system, ESR [40], utilizes the RoBERTTa neural network and demonstrated impressive F1 metric results. However, the documentation lacks details on the implementation of the *predict(...)* function. The SparseLMMS system, which incorporates large pretrained language models, offers an online demonstration and uses a contextualized mapping mechanism [41]. However, deploying this system can be intricate, and the size of its model significantly exceeds that of the model discussed in this study. Another notable system, EWISER [42], utilizes a neural supervised architecture that taps into a wealth of knowledge by embedding information from the KB graph within the neural framework, resulting in impressive F1 metric results. Nevertheless, setting up EWISER can be a challenging process.

In contrast, the system described in [43] leverages the BERT neural network and achieved a classification accuracy comparable with the results presented in this study. Unfortunately, the system's GitHub page lacks comprehensive documentation, posing challenges for its implementation on other NLP frameworks.

All reviewed works and algorithms rely on the standard WordNet dictionary, and the accompanying test dataset is referenced in related works [17,20–23]. Our solution was also evaluated on this dataset. However, this work introduces a novel component of the WSD system, which we called the "WSD debugging system". To our knowledge, no analogous approach has been presented previously. The essence of this idea, encapsulated in the title of our article ("Reversal WSD"), centers on treating the existing WSD model as the gold standard. The knowledge engineer's task is to modify the WordNet dictionary in a way that maximizes the accuracy in the standard testing dataset.

## 7. Discussion

This work introduces a WSD system, emphasizing its minimal memory footprint. The system leverages the KerasNLP library and the BERT model for efficient acquisition of embedded word vectors, promoting the conservation of computational resources. A key feature is its seamless compatibility with the Google Colab environment, enabling straightforward execution through a single "Run All" command.

Traditional construction of semantic dictionary relies on subjective interpretations by knowledge engineers for assigning semantic tags. This study proposes a novel methodology for formalizing the attribution of semantic knowledge by correlating it with neural network-derived word embedding vector clusters. Our study demonstrated the effectiveness of this approach in justifying merging of WordNet synsets where the granularity becomes excessive. While the initial validation used a moderate-sized neural network, we anticipate that further exploration with larger language models (LLMs) will produce insightful results in future investigations.

**Supplementary Materials:** The New WordNet project code and data are available online at https://github.com/aalgirdas/wordnet_onto.

**Funding:** This research received no external funding.

**Institutional Review Board Statement:** Not applicable.

**Informed Consent Statement:** Not applicable.

**Data Availability Statement:** Data are contained within the article.

**Conflicts of Interest:** The authors declare no conflicts of interest.

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
