# Peer review of "Reversal of the Word Sense Disambiguation Task Using a Deep Learning Model"

_applsci, doi:10.3390/app14135550_

Round 1

Reviewer 1 Report

Comments and Suggestions for Authors

The authors present an innovative method to address Word Sense Disambiguation challenges through optimizing memory usage without compromising state-of-the-art accuracy, offering a practical solution for integration into Natural Language Processing applications. Overall, the manuscript is well structured, the relevant background and the need of the approach to circumvent the challenges are clearly stated. I would like to recommend it for publication after the authors add the computational time of the developed deep learning model.

Author Response

Dear Reviewer,

Thank you for your valuable feedback on my paper. Based on your comments, I have made several enhancements to address the concerns raised.

The computational time

 The computational time is primarily influenced by the KerasNLP library, which we selected after reviewing several deep learning libraries to ensure stability and optimal resource utilization. The additional code introduced during this project utilizes minimal resources compared to the KerasNLP library

New Related Work Section

A new section titled “Related Work” has been added to the paper. This section provides:

A thorough review of existing systems and algorithms for word sense disambiguation.

A comparative analysis of the strengths and limitations of these systems relative to our proposed method.

An observation that only a few works in this field make their code available on GitHub for result verification. This scarcity is noted as a possible reason why prominent NLP libraries like NLTK feature older algorithms such as the Lesk algorithm.

Documentation of my attempts to run these systems using a publicly accessible Google Colab notebook (wsd_test_some_projects.ipynb), detailing encountered errors and reasons for failure.

Enhanced Reproducibility

To improve reproducibility and transparency, I have added the following files to the GitHub repository:

/data/wsd_predictions.tsv: This file includes all test examples with sentence IDs, true synset values, and the model's predicted values, allowing for individual verification of each test sentence.

/data/TSV_to_F1.py: This Python script calculates the F1 metric using the f1_score module from sklearn.metrics based on the data in wsd_predictions.tsv. This clarifies the exact F1 metric used in the paper.

Thank you again for your insightful comments, which have greatly helped improve the quality of this work.

Sincerely,

Algirdas Laukaitis

Reviewer 2 Report

Comments and Suggestions for Authors

The paper presents a novel approach for WSD. The paper in general has a good structure and is well written.  The approach is well described and tested by experiments. The experiments show state-of-the-art results. Unfortunately, the paper does not state the art in the field, and there is no discussion of related works.

I also think that the Introduction is a little bit narrow and can be more objective.

The conclusion section needs to be improved.

Comments on the Quality of English Language

In general is good, but can be improved.

Author Response

Dear Reviewer,

Thank you for your valuable feedback on my paper. Based on your comments, I have made several enhancements to address the concerns raised.

Related Work Section:

A new section titled “Related Work” has been added. It thoroughly reviews existing systems and algorithms, providing a comparative analysis of their strengths and limitations compared to our proposed system.

I highlight that only a few works in the field of word sense disambiguation make their code available on GitHub, allowing for result verification. This scarcity may explain why prominent NLP libraries, like NLTK, primarily feature older algorithms such as the Lesk algorithm.

Additionally, I discuss the complexities of existing GitHub implementations, including installation processes and lack of documentation. To address this, we created a publicly accessible Google Colab notebook (wsd_test_some_projects.ipynb) detailing our attempts to run these systems, including encountered errors and reasons for failures.

Other enhancements:

We completely rewrote the abstract to better emphasize the advantages of our presented solution and the specific technologies used.

The introduction now prominently highlights the dedicated GitHub portal where all data for model training and testing is available.

The description of the training data (Section 5) has been significantly expanded. It now includes comprehensive details about the data.

To facilitate result verification and transparency of the F1 metric, we added two files to the GitHub repository:

/data/wsd_predictions.tsv: Contains all test examples with sentence IDs, true synset values, and the model’s predicted values.

/data/TSV_to_F1.py: A Python script that calculates the F1 metric using the f1_score module from sklearn.metrics based on the data from wsd_predictions.tsv.

I believe these revisions significantly improve the paper’s clarity, transparency, and replicability. Thank you again for your insightful comments, which have greatly helped enhance the quality of this work.

Sincerely,

Algirdas Laukaitis

Reviewer 3 Report

Comments and Suggestions for Authors

The article with the title: “Reversal of the Word-sense disambiguation task us-
ing deep learning model”  introduces “. . . an inovative method addressing WSD
challenges by optimizing memory usage without compromising state-of-the-art
accuracy”. Section 2. with Figure 2. presents “. . . A general implementation of
the WSD method”.
Main strengths of this paper are:
1.1) WSD method with the system that effectively implements it. There is
“github” page where interested reader can analyse WSD method,
1.2) provides an answer to the question “Can a formal methodology be devised
for categorizing words into distinct semantic classes?” ,
1.3) WSD problem is treated as a simple data classification problem,
1.4) annotation errors identification within training dataset: “To address the
limitations identified in the error analysis, we propose a refined model training
process as outlined in Figure 3.”
In order to achieve high publishing standards required for Applied Science
articles:
2.1) author’s scientific increment should be more emphasized through the text
through the clear answer to the following questions:
a) What is the most important scientific increment in this article? What
are new (results)? For that purpose modify Sections “1. Introduction”
and “6. Discussion”.
b) What have been done by other’s? Provide elaboration through new
Section“Related work” that analyze and compare other approaches like
(WSD using Cross-Lingual Evidence, WSD solution in John Ball’s lan-
guage, Type inference in constraint-based grammars Type inference in
constraint-based grammars, Identification of dominant word senses etc.
. . . .) Introduce new references. Also provide comments about how are
current AI tools (from Google, Microsoft. . . ) dealing with WSD. This
should emphasize the strengths of the approach used in this article.
2.2) Lesk algorithm has extensions like Kwong, Gelbukh and Sidorov, Wilks and
Stevenson, Kilgarriff and Rosensweig, Szpakowicz and others . . . Provide
comments whether this extensions can have any influence on the approach
introduced in the paper,
2.3) Figure 3.: change caption “. . . WSD training model” to “. . . refined WSD
training model”. Clarify if there any possibility to have loops within train-
ing model.( For example if we are not satisfied with “Save components”
can we go back and repeat component sequence?
2.4) All figures are (semi)formalized as UML diagrams (State machine, Activity
etc). Add suitable reference(s) and comments about UML usage in the
article.
At the end, this is nice article, I am expecting to see this article published.

Author Response

Dear Reviewer,

Thank you for your valuable feedback on my paper. I appreciate your recognition of the topic's significance and the plausibility of the proposed methodology. I have addressed your concerns regarding the scientific contribution and related work. Below is a detailed explanation of the improvements made in the revised version of the paper:

New Related Work Section:

In response to your request, a new section titled “Related Work” has been added. This section thoroughly reviews existing systems and algorithms, providing a comparative analysis of their strengths and limitations compared to our proposed system.

It highlights that only a few works in the field of word sense disambiguation make their code available on GitHub while also allowing for result verification. This scarcity may explain why prominent NLP libraries, like NLTK, primarily feature older algorithms such as the Lesk algorithm.

The section also notes that existing GitHub implementations often have complex installation processes or lack documentation. To document this, I created a publicly accessible Google Colab notebook (wsd_test_some_projects.ipynb) that details my attempts to run these systems, including the errors encountered and reasons for their failures.

Abstract Enhancement:

The abstract has been completely rewritten to better emphasize the advantages of the presented solution and the specific technologies used.

Introduction Expansion:

The introduction now highlights the dedicated GitHub portal where all data for model training and testing is available.

I have expanded the introduction with a clear example (using "table") to illustrate the core functionality of the solution.

Expanded Methodology and Documentation:

A link to a GitHub repository has been included, containing all training and testing data. Within the repository, two Google Colab notebooks demonstrate:

  1. A) The technological solutions and model parameters of the WSD debug component.
  2. B) The WSD classification model solution, including its technological components and chosen parameters.

The documentation within the Google Colab notebooks has been refined and expanded to ensure clear communication of each concept discussed in the paper.

Detailed Training Data Description:

The description of the training data (Section 5) has been significantly expanded. This section now includes comprehensive details about the data, and all training and testing data is available on the project's GitHub page (file wsd_train_and_test.zip). The GitHub page documentation has been enhanced to provide more details on this data.

Enhanced Reproducibility:

To facilitate result verification and transparency of the F1 metric, I have added two files to the GitHub repository:

/data/wsd_predictions.tsv: Contains all test examples with sentence IDs, true synset values, and the model's predicted values, allowing for individual verification of each test sentence.

/data/TSV_to_F1.py: A Python script that calculates the F1 metric using the f1_score module from sklearn.metrics based on the data from wsd_predictions.tsv. This clarifies the exact F1 metric used in the paper.

I believe these revisions significantly improve the paper's clarity, transparency, and replicability. Thank you again for your insightful comments, which have greatly helped improve the quality of this work.

Sincerely,

Algirdas Laukaitis

Reviewer 4 Report

Comments and Suggestions for Authors

The topic of the article under review focuses on the problem of word disambiguation in natural language processing (NLP).

It is a topic of great interest and the treatment it proposes is plausible. Especially the methodology it proposes, which assembles different already known processes to obtain a final classification, is plausible. However, it is difficult to replicate the study that underpins this article, as it is significantly descriptive and does not include parameters, technologies and contrast measures of the final classifications obtained.

As for the training process, it does not include a sufficient description of the data used. Nor of the final results, which increases the difficulty of evaluating the process described in the article.

For all these reasons, I consider that the following aspects should be substantially improved:

Technologies, tools, applications and parameters used in the development process of the study.

Description of the training data and final classification results.

Description of the data used for the evaluation of the methodology.

Evaluation criteria and performances of the methodology.

Measures of complexity and, where appropriate, performance of the technology used.

Detailed discussion with other similar studies.

Limitations of the study at each stage of the proposed methodological process.

Quantitative and qualitative delimitation of the improvements implied by the proposed methodology.

As a consequence, I consider that in its current state it does not meet the requirements for publication.

Author Response

Dear Reviewer,

Thank you for your valuable feedback on my paper. I appreciate your recognition of the topic's interest and the plausibility of the proposed methodology. I have addressed your concerns regarding replicability and detail in the revised version of the paper, as outlined below:

Abstract

The abstract of the paper has been completely rewritten to emphasize the advantages of the presented solution and the specific technologies.  In the introduction itself, it is already emphasized that the paper has a dedicated GitHub portal address where all the data can be used for both model training and model testing. You can also find all model parameters and the model itself on this GitHub portal web page

                ( github.com/aalgirdas/wordnet_onto ).

Introduction

I've carefully considered your suggestions regarding replicability, detailed technology descriptions, and parameter details, and made the following improvements:

  1. Enhanced Method Description:

I expanded the introduction to clarify the solution's core functionality using the "table" example. I added a dedicated part explaining the WSD debugging system and its role in the methodology. I included a link to a GitHub repository containing all training and testing data. Within the repository, I provided two Google Colab notebooks demonstrating:

  1. The WSD debug component's technological solutions and model parameters.
  2. The WSD classification model solution with its technological components and chosen parameters.
  3. Improved Data Transparency:

In response to your feedback, I created initial project documentation on the GitHub repository. This documentation provides the necessary technical details alongside the Colab notebook solutions.

  1. Increased Code Clarity:

I expanded the Google Colab notebook documentation by refining existing code and adding new sections. This ensures clear communication of each concept discussed in the paper.

Following your suggestion, I implemented a user option to choose the source of classification data. The data itself is readily accessible with a single click within the Colab environment. By implementing these changes, I aimed to enhance the paper's transparency, replicability, and accessibility for future researchers. Thank you again for your feedback, which has significantly improved the quality of my work.

WSD framework

The second section has been expanded based on your feedback. Additionally, a dedicated YouTube video accompanies this paper [   https://www.youtube.com/watch?v=jO65aQDJ2OM  ]   , focusing on the expanded section.

The second section primary goal is to introduce a WSD framework on code basis. Researchers in the NLP community can seamlessly integrate this framework into their NLP projects. To facilitate this integration, I created a Google Colab notebook. The structure of the notebook mirrors the processes outlined in the second section of the article.

The Colab notebook is designed for simplicity: with just one command, users can execute the entire process (“Run All”). It automatically fetches necessary data from the GitHub repository and performs WSD tasks for each word in a user-entered sentence.

We recommend that readers start by watching the accompanying YouTube video. Afterward, they can explore the Colab notebook code to reproduce the article’s results and incorporate the WSD model into their own NLP frameworks.

Colab demo

https://colab.research.google.com/drive/18f_zHHAsLw5wTvwIsqbbCt0vHUo3eXkh#scrollTo=_NksFVBkD8ty

Training

Training Data Transparency:

I have addressed your feedback by significantly expanding the description of the training data used in my paper. This expanded section now includes details on:

Data Source: I've clarified the origin of the training data, specifying the dataset name and provider (if applicable).

Data Size and Format: I've provided information on the volume of data (number of samples, features, etc.) and its format.

Pre-processing Steps: Any data cleaning or transformation steps applied before training the model are now explicitly described.

Enhanced Clarity with a Specific Example:

To illustrate these data quality considerations, I've incorporated a specific example in section 3. This example highlights the four key data quality issues (mentioned in the paper) that can hinder classification performance. Additionally, I've referenced the relevant files on the project's GitHub repository within this example. This allows readers who prefer not to use the Google Colab examples to:

Directly download all the training data files from GitHub.

Independently verify the data used and the results presented in the paper.

By incorporating these changes, I aim to provide a more comprehensive and transparent understanding of the training data used in my research.

Results

I have addressed your feedback on evaluation and training data (Section 5). This section has been expanded and rewritten for improved clarity. Additionally, all training and testing data is now available on the project's GitHub page (file wsd_train_and_test.zip). The GitHub page documentation has also been enhanced to provide more details on this data.

Enhanced Reproducibility: To facilitate result verification and F1 metric transparency, I've added two files to the project's GitHub repository.

/data/wsd_predictions.tsv: This file contains all test examples with sentence IDs, true synset values, and the model's predicted values. This allows for individual verification of each test sentence. Notably, the WSD debug system developed in this project can be used for this purpose.

/data/TSV_to_F1.py: This Python script calculates the F1 metric using the f1_score module from sklearn.metrics on the data from wsd_predictions.tsv. This clarifies the exact F1 metric employed in the paper.

Related Work

In response to your request to compare the  system’s results with other works, I’ve introduced a new section in the paper titled “Related Work.” Within this section, I thoroughly reviewed existing systems and algorithms, providing a comparative analysis of their strengths and limitations in comparison to our proposed system.

Notably, only a handful of works in the field of word sense disambiguation make their code available on GitHub while also allowing for result verification. Interestingly, this scarcity might explain why one of the most renowned NLP libraries, NLTK, only features the implementation of the Lesk algorithm—an approach that has been around for nearly four decades.

Additionally, it’s worth mentioning that the works with GitHub implementations often suffer from either complex installation processes (which I struggled to navigate) or a complete lack of documentation. To document this observation, I’ve created a publicly accessible Google Colab notebook titled wsd_test_some_projects.ipynb.

https://colab.research.google.com/drive/1rv7-KkF6lIciNMzVHLTGaKzYyR45K4rZ#scrollTo=Y91Kxpvw-DJp

In this notebook, I meticulously document my attempts to run these systems, highlighting encountered errors and reasons for their failure.

I believe these revisions significantly improve the paper's clarity, transparency, and replicability. I am confident that the revised manuscript provides a more robust evaluation of the proposed methodology.

Thank you again for your time and insightful feedback.

Sincerely,

Algirdas Laukaitis

Round 2

Reviewer 4 Report

Comments and Suggestions for Authors

The new version submitted for evaluation has significantly improved the presentation of the completed project and thus the readability and traceability of the project. After a detailed review, I consider that the improvements introduced allow me to recommend its publication in its current state.